# An Evaluation of Non-Communicable Diseases and Risk Factors Associated with COVID-19 Disease Severity in Dubai, United Arab Emirates: An Observational Retrospective Study

**DOI:** 10.3390/ijerph192114381

**Published:** 2022-11-03

**Authors:** Najwa Abdulwahed Al Bastaki, Mohamud Sheek-Hussein, Ankita Shukla, Najlaa Al-Bluwi, Basema Saddik

**Affiliations:** 1Department of Public Health Protection, Dubai Health Authority, Dubai P.O. Box 4545, United Arab Emirates; 2Institute of Public Health, College of Medicine and Health Sciences, United Arab Emirates University, Al-Ain P.O. Box 15551, United Arab Emirates; 3School of Public Health, Loma Linda University, Loma Linda, CA 92354, USA; 4Sharjah Institute for Medical and Health Sciences Research, University of Sharjah, Sharjah P.O. Box 27272, United Arab Emirates; 5Department of Family and Community Medicine and Behavioural Sciences, College of Medicine, University of Sharjah, Sharjah P.O. Box 27272, United Arab Emirates

**Keywords:** COVID-19, severity, comorbidities, prevalence, predictors

## Abstract

The aim of this study was to determine the prevalence of non-communicable diseases and their correlation with COVID-19 disease severity among patients infected in Dubai. Clinical and demographic data were extracted from hospital records of 34,687 COVID-19 patients who visited or were admitted into Dubai hospitals between 28 January 2020 and 30 September 2020. Prevalence of co-morbidities in COVID-19 patients were assessed. The main risk factors associated with COVID-19 disease severity were also identified using three regression models. All co-morbidities were significantly associated with COVID-19 severity in the bivariate analysis (*p*-value ≤ 0.05) except for vitamin-D deficiency and chronic lower respiratory diseases. Patients with ischemic heart diseases (AOR: 2.08; 95% CI: 1.37, 3.15), pulmonary and other heart diseases (AOR: 2.13; 95% CI: 1.36, 3.32) and chronic kidney diseases (AOR: 1.81; 95% CI: 1.01, 3.25) had higher odds of severe COVID-19 symptoms. Suffering from multiple co-morbidities increased the odds of developing severe COVID-19 symptoms substantially in comparison to having only one co-morbidity i.e., (AOR: 1.52; 95% CI 1.76–2.60) to (AOR: 2.33; 95% CI: 1.37, 3.97). Identifying these risk factors could assist in the early recognition of high-risk populations and ensure the most appropriate preventive measures and required medical management during the pandemic.

## 1. Introduction

Since its first detection in Wuhan, China in December 2019, the severe acute respiratory syndrome coronavirus 2 (SARS-CoV-2), later known as coronavirus disease 2019 (COVID-19) has spread worldwide to over 223 countries and territories including the United Arab Emirates (UAE) [1]. In the UAE there have been over 892,000 confirmed COVID-19 cases and 2302 total deaths as of 3rd April 2022 [1]. Research during the early phases of the pandemic indicates that demographic and behavioral factors such as sex, age and smoking may impact the severity of infections with COVID-19 [2,3,4,5,6].

Diabetes, hypertension and coronary heart disease have been reported as the most common comorbidities among COVID-19 patients [5,7,8,9]. Several studies, including those conducted in the Middle East and North Africa (MENA) region, have demonstrated that COVID-19 severity and clinical outcomes are worse for patients with pre-existing comorbidities such as pulmonary or kidney disease [4,5,10], cancer [11], nervous system diseases [12,13], diabetes [5,14,15], and hypertension or coronary heart disease [8,16]. Moreover, it has been reported that patients with multiple comorbidities experience worse COVID-19 symptoms and even mortality [17].

Among the total 9.89 million people living in the UAE in 2020, the majority (88%) were young adult expatriates [18]. However, the burden of diseases in the UAE is heavily dominated by non-communicable diseases (NCDs) with the UAE reporting the highest relative prevalence of diabetes in the MENA region (16.3% among adults aged 20 to 79 years) [19], and obesity prevalence of 32.3% among expatriate adults [20]. The positive association between NCDs and COVID-19 severity around the world combined with the high rate of NCDs in the UAE, places the country at higher risk of disease infection and worse clinical outcomes. Studies from Dubai and Al-Ain have demonstrated that patients with diabetes, chronic kidney disease, and vitamin-D Deficiency have more severe COVID-19 symptoms and mortality [21,22].

The World Health Organization (WHO) recently reported that pre-existing NCDs may increase COVID-19 severity [23]. Therefore, identifying patients at highest risk is crucial to improving health outcomes. However, studies from the UAE that have examined the association between pre-existing comorbidities and COVID-19 severity have been limited to small samples and single comorbidities. Therefore, the current study aims to fill this research gap by using a large sample of medical records from the Dubai Health Authority (DHA) to examine the prevalence of comorbidities among COVID-19 patients and the risk associated with existing comorbidities in causing severe COVID-19 symptoms. Furthermore, this study also examines the impact of multiple comorbidities on the severity of COVID-19 patient outcomes as compared to those with a single comorbidity or no comorbidity.

## 2. Methods

### 2.1. Study Design and Setting

We conducted a retrospective observational study using confirmed real-time polymerase chain reaction (RT-PCR) positive COVID-19 cases in the Emirate of Dubai, UAE. Data were extracted from the DHA health information system section. DHA is the governmental health authority responsible for providing medical services to the population of Dubai and governs four hospitals comprising approximately 1500 beds for adults and one 200 bed hospital for children. The EPIC electronic health record system is utilized throughout DHA facilities to document all medical information related to the patient’s hospital admission including patient information, symptoms, treatments, medical history and all relevant clinical data.

### 2.2. Participants and Study Size

Data from a total of 57,292 patients of all ages and both sexes that were clinically diagnosed with COVID-19 and had tested positive in PCR laboratories for hospitalization, or were in isolation in Dubai DHA government hospitals, or quarantine centers under DHA between 28th January 2020 to 30th September 2020 were extracted from the DHA health information system. Inclusion criteria included, (i) adults aged 20 years and above, (ii) COVID-19 positive test results that met the case definition, and (iii) visited a DHA facility during the study period. A total of 34,687 patient data that fulfilled these criteria were included in the bivariate analysis. A detailed description and breakdown of included, missing, and repeated cases are presented in Figure 1.

### 2.3. Defining Dataset Variables, Co-Morbidities, and Disease Severity

The dataset included: demographic variables (age, sex, residence, and smoking status); Body Mass Index (BMI) categorized as: underweight (<18.5), normal (18.5–24.9), overweight (25.0–29.9), and obese (>29.9) [24]; and pre-existing co-morbidities (grouped into 14 broad categories as per the International Classification of Diseases, Tenth Revision (ICD-10) (Table A1). Vitamin-D deficiency was classified as a serum 25-hydroxyvitamin D (25OHD) value less than 30 nmol/L as defined by the Institute of Medicine [25]. There were very few cases of tuberculosis and viral hepatitis, so these were excluded from further statistical analyses.

COVID-19 severity was classified into four categories following the WHO ISARIC classification [26] as; (i) Mild: symptomatic patients meeting the case definition for COVID-19 without evidence of viral pneumonia or hypoxia; (ii) Moderate: A Patients with clinical signs of pneumonia (fever, cough, dyspnea, fast breathing) but no signs of severe pneumonia, including SpO2 ≥ 90% in room air; (iii) Severe: A Patients with clinical signs of pneumonia (fever, cough, dyspnea, fast breathing) plus one of the following: respiratory rate > 30 breaths/min, severe respiratory distress, or SpO2 < 90% on room air); and (iv) Critical: Individuals who have respiratory failure, septic shock, and/or multiple organ dysfunction.

### 2.4. Statistical Analyses

Data were analyzed using Stata (16) [27]. Descriptive data are displayed in counts and percentages. Pearson Chi-Square (*χ*^2^) analysis identified associations between comorbidities and COVID-19 severity. Only those comorbidities that showed significant association in Chi-square analysis were further included in the binomial logistic regression analysis. COVID-19 severity was recoded into two groups for the regression model with mild and moderate severity classified as non-severe and severe and critical classified as severe. Three models were used for the multivariable logistic regression to identify predictors of COVID-19 severity. The first model controlled for comorbidity, the second model controlled for both comorbidity and sociodemographic characteristics, while the third model controlled for comorbidity, sociodemographic characteristics, and number of co-morbidities. The results are presented as adjusted odds ratios (aORs) with 95% confidence intervals (CI) after adjustment for the effects of potential confounders. A two-tailed *p* < 0.05 was considered statistically significant.

### 2.5. Ethical Approval

The study was approved by the DHA research ethics committee (DSREC-07/2020_18) on 22 July 2020 and the Emirates Institutional Review Board for COVID-19 Research DOH/CVDC/2020/1576 on 9 August 2020. Both committees waived the need for obtaining patient consent.

## 3. Results

The majority of COVID-19 patients were male (80%), aged 30–39 years (37%), and expatriates (90%) (Table 1). Almost two thirds of COVID-19 patients were either overweight (41%) or obese (25%) while 3% were current smokers and 1% former smokers. Most patients had mild COVID-19 severity (92%) followed by moderate (5%), and only 2% of patients were classified as severe and admitted to ICU (Table 1).

A higher percentage of elderly, i.e., aged 60 years and above (7.2%) and male patients (2.1%) were admitted into the ICU (Figure 2). Also, patients who did not maintain normal weight (i.e., underweight (23%), obese (13.4%), or overweight (16.2%)) were more likely to be admitted into the ICU.

The most prevalent pre-existing comorbidities among COVID-19 patients were diabetes (5.27%) and hypertensive disorders (4.40%) followed by vitamin-D deficiency (2.78%), nervous system diseases (1.2%) and thyroid gland disorders (1.07%). Less than 1% of patients had other co-morbidities as shown in Figure 3.

A significantly higher percentage of patients who had pulmonary and other heart diseases (27.64%), chronic kidney disease (27.08%), cerebrovascular diseases (25.95%), ischemic heart diseases (19.57%), and hypertensive disorders (7.87%) were reported as being critical COVID-19 cases (Table 2). There was significant association found between the number of existing co-morbidities and COVID-19 severity with patients reporting multiple co-morbidities having worse (critical) severity (11.46%).

The Chi-square test for association between comorbidities and COVID-19 severity for the 4220 patients included in the multivariable regression analysis are displayed in Table A2. Table 3 presents results from the multivariable logistic regression analysis. Among the three regression models, the Pseudo R2 values according to (McFadden estimates) were 0.062 for model one, 0.087 for model two, and 0.090 for model three, respectively, indicating that model three provided the best goodness of fit to the data.

In the first model, patients with diabetes (aOR: 2.32; 95% CI: 1.87, 2.88), ischemic heart diseases (aOR: 3.23; 95% CI: 2.19, 4.76), pulmonary and other heart diseases (aOR: 2.86; 95% CI: 1.88, 4.35), and chronic kidney diseases (aOR: 1.98; 95% CI: 1.10, 3.57) had higher odds of developing severe COVID-19 infection (Table 3; Model 1). However, once individual level characteristics such as age, gender, BMI, and smoking status were controlled for in the model, thyroid gland and mental and behavioral disorders also became significant predictors of worse COVID-19 severity. (Table 3; Model 2). In model 2, age, gender, BMI, and nationality were statistically significant predictors of worse COVID-19 severity. Increasing age was associated with increased likelihood of worse COVID-19 severity; 40–49 years (aOR: 2.04; 95% CI: 1.45, 2.86), 50–59 years (aOR: 2.47; 95% CI: 1.73, 3.52), 60 years and over (aOR: 3.28; 95% CI: 2.24, 4.80), and females had lower odds of developing severe COVID-19 infection compared to their male counterparts (aOR: 0.70; 95% CI: 0.55, 0.89). Patients who were underweight (aOR: 2.24; 95% CI: 1.18, 4.23); overweight (aOR: 1.39; 95% CI: 1.13, 1.72) or obese (aOR: 1.58; 95% CI: 1.25, 1.99) had higher odds of severe COVID-19 as compared to those within normal weight. After controlling for the number of comorbidities in the final model (Table 3; Model 3) the significance for diabetes, thyroid gland disorders and mental and behavioral disorders disappeared. Patients who had a pre-existing comorbidity, whether a single co-morbidity (aOR: 1.52; 95% CI: 1.13, 2.03) or multiple comorbidities (aOR: 2.33; 95% CI: 1.37, 3.97) had higher odds of developing severe COVID-19 infection as compared to those with no existing comorbidities.

## 4. Discussion

Almost two years have passed since the COVID-19 outbreak was declared a pandemic [28], and research continues to advance knowledge of disease prognosis, treatment and the long-term effects. Although COVID-19 treatment protocols have been proposed since the early stages of the pandemic [29,30,31,32], none have been widely adopted. In Dubai, hospitals are accredited by the Joint Commission International (JCI) accreditation body and JCI patient standards of care were maintained throughout the pandemic. However, in the absence of specific treatment protocols and the emergence of new variants, the identification of high-risk patients prone to clinical deterioration is critical for ensuring adequate and timely access to intensive care treatment and resource allocation.

The results from this study demonstrate that although most patients who were admitted to DHA facilities had mild COVID-19 severity, a significant number had moderate-to-critical health outcomes and required ICU admission. These results are consistent with prior reports concerning COVID-19 infections in China [33], which reported severe (14%) and critical (5%) symptoms and more recently in Abu Dhabi, UAE with similar reports of severe (1.8%) and critical (0.5%) outcomes [34]. COVID-19 infection was also observed to be highest among men and adults aged 20–50 years. Researchers from Abu Dhabi (UAE) who reported similar findings in 2021, indicated that one of the factors supporting these findings is that most expatriate men are of working age (20–50 years). This population is largely employed in high-risk occupations such as construction, hospitality, domestic work, and retail, and they live in shared accommodation where infections are quite likely to occur [35,36].

Compared with younger adults, older patients aged 60 years and over were more likely to experience severe COVID-19 symptoms. Several studies have shown similar results [6,37]. Interestingly, this association has also been observed in previous respiratory pandemics such as influenza (H7N9) [38,39]. Ageing as a risk factor for COVID-19 severity is particularly important given the rapid increase in the number and proportion of older people globally. The United Nations estimates that by 2050, one in six people worldwide will be 65 years of age or older, compared to one in eleven in 2019 [40]. Additionally, older patients with infections often present with atypical symptoms, making the diagnosis and treatment challenging [41]. With a median age of 30.3 years reported for Dubai [18], the UAE has a relatively young population. The disease trends observed in our study may be explained by these factors.

Another interesting finding in our study was the effect of weight on COVID-19 severity. Underweight patients had a 2.24 times higher likelihood of experiencing severe health outcomes than normal weight patients. Furthermore, obese patients had a 58% higher risk of experiencing poorer COVID-19 health outcomes. Studies from the MENA region have reported similar results including Kuwait [42], the Kingdom of Saudi Arabia (KSA) [5], and Iran [43]. In fact, the study of interactions between obesity and infectious respiratory diseases began even before the COVID-19 pandemic. Earlier research has shown that obesity can restrict ventilation and damage the immune system, both of which lead to an increased susceptibility to severe respiratory illness and adverse outcomes [44,45,46]. The exact mechanism by which obesity contributes to severe outcomes for COVID-19 is still unclear; however, one possible explanation might be the increased expression of the functional angiotensin-converting enzyme 2 (ACE2) receptor for SARS-CoV2 in obese individuals [47]. Further, obesity is associated with dysregulated lipid synthesis and clearance which results in increased pulmonary inflammation and injury [47,48]. The high prevalence of overweight and obesity in the UAE and other countries of the Gulf Cooperation Council (GCC) [20,49], emphasizes the need for future interventions and policies aimed at improving the health of underweight and obese patients.

Similar to other studies from the UAE, we also found that diabetes was the most common comorbidity among COVID-19 patients [34,50]. Higher mortality and worse disease outcomes have previously been reported in patients with diabetes [5,51]. Given the high prevalence of diabetes in the UAE in comparison to other countries [16,19], it is of critical importance to highlight the impact of COVID-19 on diabetes. Diabetes is known to have a negative impact on the innate immune response through insulin resistance and β-cell damage. Additionally, diabetes may impair the immune response by impairing macrophage and lymphocyte function, thus making patients more susceptible to disease complications and poor outcomes [52]. Based on our current observations and the literature cited previously, diabetes continues to be a major risk factor for poor prognosis among patients with COVID-19.

Although a relatively small percentage of patients in our study suffered from ischemic, pulmonary, and other heart diseases, those who did were more likely to develop severe COVID-19 outcomes, consistent with the literature [51,53]. The risk associated with COVID-19 severity among cardiac patients is relevant because cardiovascular patients are at an increased risk of cell damage due to direct viral damage, systemic inflammatory responses, destabilized coronary plaques, and aggravated hypoxia [54]. Moreover, cardiovascular patients often have increased ACE2 receptor expression, which facilitates the entry of viruses into the host body [53].

Chronic kidney disease has long been reported as a major public health problem and a significant risk factor for poor health outcomes [55]. Our finding that chronic kidney disease is a significant predictor of COVID-19 severity is consistent with previous observations in Spain and Mexico where studies found that kidney disease is significantly associated with severe COVID-19 symptoms and higher mortality compared with those without chronic kidney disorders [56,57]. Chronic kidney disease has become more prevalent over the past decade, with 13.4% of the population affected worldwide [58]. Among UAE nationals, the prevalence of kidney disease has been estimated to range from 2.8% to 4.6% [59]. These results may be underestimated due to the high prevalence of known kidney disease risk factors such as smoking and hypertension in the country [60,61]. There are several mechanisms responsible for kidney involvement during COVID-19 infection including cytokine damage (an exaggerated immune response), organ crosstalk (a mutual biological communication between distant organs mediated by signaling factors) and systemic effects (fluid expansion and hemodynamic instability) [62,63].

Our study also found that COVID-19 severity is strongly influenced by the presence of multiple comorbidities such that once these comorbidities were controlled in the model, the effects of diabetes, thyroid disorders, mental disorders, and behavioral disorders no longer appeared. These findings suggest that an increase in comorbidities is a significant predictor of COVID-19 severity, ICU admission, ventilator assistance, and even mortality-consistent with previous studies [64,65]. Additionally, multiple comorbidities are typically seen in older individuals. Therefore, aging and multiple comorbidities increase the likelihood of developing critical complications and outcomes from a novel coronavirus disease.

This study is particularly noteworthy for its comprehensive coverage of all DHA hospitals in Dubai and a large sample size of 34,687 patients collected over a period of nine months. The study provides further evidence to support the previous worldwide literature concerning the impact of age and pre-existing comorbidities on COVID-19 severity. Moreover, this data is from the first wave of the pandemic, which has two advantages. The first is that these results are not confounded by vaccination status, since vaccines were not available during 2020 [66] and secondly, these results are not confounded by serious second COVID-19 reinfection, because natural immunity held up throughout 2020. Nevertheless, one disadvantage of the first wave data is that the virus has since mutated substantially, so there can be some quantitative variability in the reported associations for the viral strains that are now circulating during 2022. However, qualitatively speaking, the associations can be reasonably expected to persist as reported. It is also important to note that the present study can only examine associations and cannot infer causality since it is retrospective. Additionally, the prevalence of NCDs such as diabetes, hypertension and obesity in the present study was lower than in previous studies conducted in the UAE. This may be due to the majority of our study population being relatively young and from Dubai, which comprises mostly expatriate males aged 20 to 40 years old. Moreover, since the comorbidity data were obtained from hospital records during admission, it is possible that some comorbidities were not detected and underreported.

## 5. Conclusions

The results of this study, in which data were extracted from a large dataset of patient medical records collected during the peak of the COVID-19 outbreak in Dubai, demonstrate that obese patients, those suffering from ischemic, pulmonary heart and kidney diseases and those afflicted with multiple co-morbidities are more likely to experience worse COVID-19 outcomes. Continually monitoring COVID-19 patients with underlying comorbidities will reduce the risk of severe outcomes. Due to the nature of the pandemic and the possibility of new variants emerging, it is imperative to improve healthcare providers’ awareness of vulnerable patients including expatriates and the elderly, so that specific strategies for their management of COVID-19 can be developed. Understanding the risk factors associated with COVID-19 infection will facilitate the identification of those at high risk, prompt their referral to hospitals at the earliest onset of symptoms, and prevent COVID-19 from progressing to severe levels. Furthermore, we recommend enhancing the systematic and real-time availability of epidemiological data for future more precise longitudinal and interventional studies.

## Figures and Tables

**Figure 1 ijerph-19-14381-f001:**
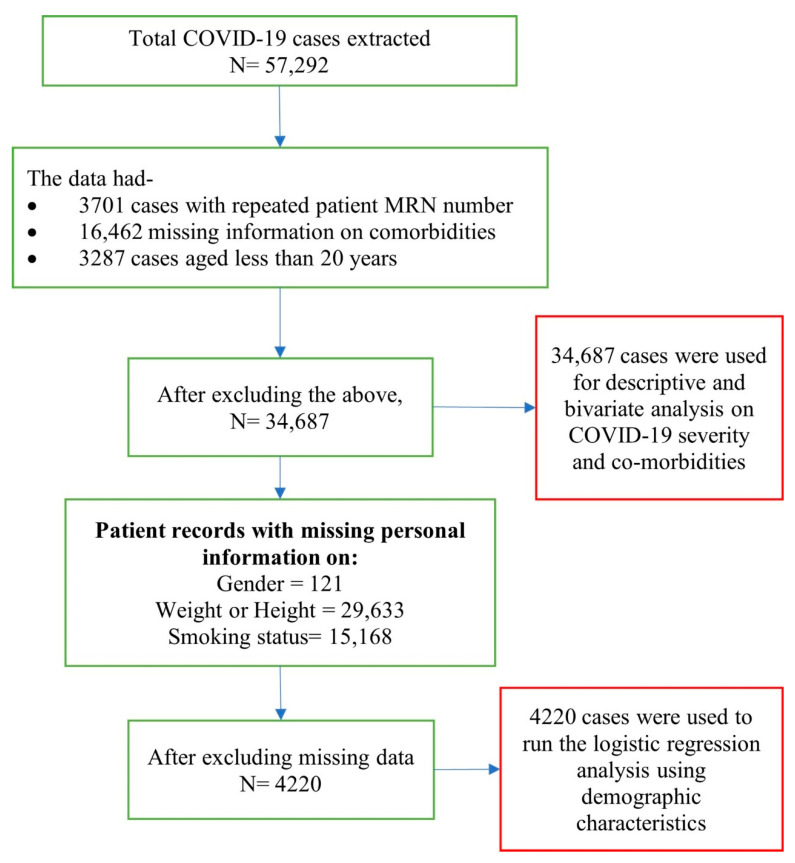
Description and breakdown of patient data.

**Figure 2 ijerph-19-14381-f002:**
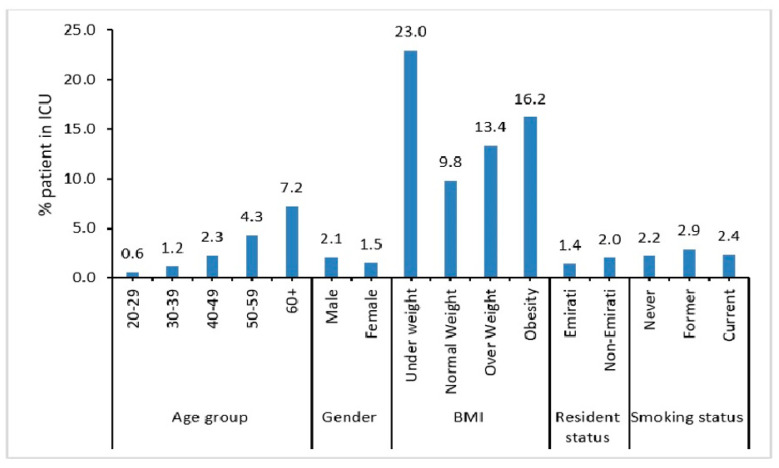
Demographic characteristics of COVID-19 patients admitted to ICU.

**Figure 3 ijerph-19-14381-f003:**
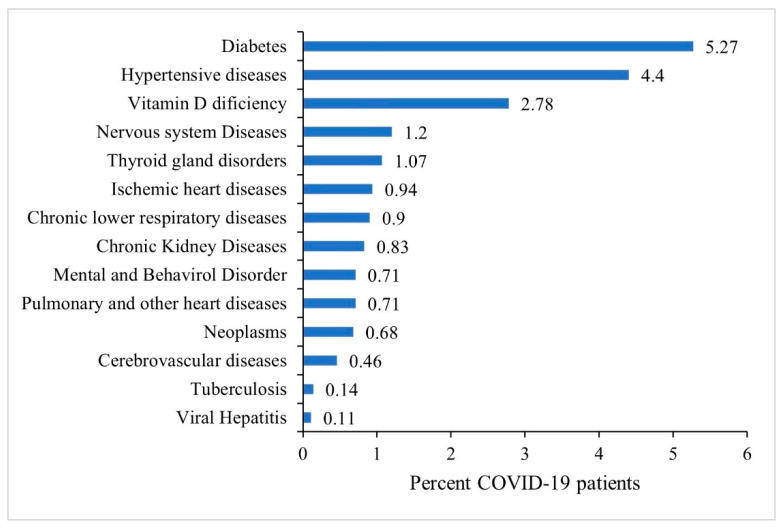
Prevalence of existing co-morbidities among all COVID-19 patients, N = 34,687.

**Table 1 ijerph-19-14381-t001:** Sociodemographic and health characteristics of the study population.

Characteristics	N	%
Age group		
20–29	7962	22.95
30–39	12,874	37.12
40–49	8011	23.1
50–59	4031	11.62
60+	1808	5.21
Sex		
Male	27,643	79.97
Female	6923	20.03
BMI		
Underweight	87	1.72
Normal Weight	1636	32.37
Overweight	2074	41.04
Obesity	1257	24.87
Nationality		
Emirati	3365	9.7
Non-Emirati	31,322	90.3
Smoking status		
Never	18,726	95.94
Former	204	1.05
Current	589	3.02
COVID-19 severity		
Mild	31,757	91.55
Moderate	1847	5.32
Severe	399	1.15
Critical	684	1.97
ICU admission		
No	34,003	98.03
Yes	684	1.97

**Table 2 ijerph-19-14381-t002:** Chi-square test for association between comorbidities and COVID-19 severity, N = 34,687.

	Mild		Moderate		Severe		Critical		Total	χ^2^ Value	*p*-Value
	n	%	n	%	n	%	n	%	N		
Vitamin D deficiency (<30 nmol/L)											
No	30,863	91.52	1794	5.32	394	1.17	671	1.99	33,722	5.59	0.133
Yes	894	92.64	53	5.49	5	0.52	13	1.35	965		
Chronic lower respiratory diseases											
No	31,494	91.62	1822	5.3	393	1.14	667	1.94	34,376	27.1	≤0.001
Yes	263	84.57	25	8.04	6	1.93	17	5.47	311		
Diabetes											
No	30,587	93.08	1479	4.5	292	0.89	502	1.53	32,860	1950.4	≤0.001
Yes	1170	64.04	368	20.14	107	5.86	182	9.96	1827		
Thyroid gland disorders											
No	31,450	91.65	1821	5.31	389	1.13	657	1.91	34,317	66.7	≤0.001
Yes	307	82.97	26	7.03	10	2.7	27	7.3	370		
Hypertensive disorders											
No	30,678	92.51	1594	4.81	326	0.98	564	1.7	33,162	920.2	≤0.001
Yes	1079	70.75	253	16.59	73	4.79	120	7.87	1525		
Ischemic heart diseases											
No	31,562	91.86	1802	5.24	376	1.09	620	1.8	34,360	698.8	≤0.001
Yes	195	59.63	45	13.76	23	7.03	64	19.57	327		
Pulmonary and other heart diseases											
No	31,629	91.84	1810	5.26	386	1.12	616	1.79	34,441	950.9	≤0.001
Yes	128	52.03	37	15.04	13	5.28	68	27.64	246		
Cerebrovascular diseases											
No	31,666	91.71	1826	5.29	394	1.14	643	1.86	34,529	507.3	≤0.001
Yes	91	57.59	21	13.29	5	3.16	41	25.95	158		
Chronic kidney diseases											
No	31,692	91.75	1814	5.25	392	1.13	645	1.87	34,543	597.8	≤0.001
Yes	65	45.14	33	22.92	7	4.86	39	27.08	144		
Neoplasms											
No	31,565	91.62	1823	5.29	393	1.14	670	1.94	34,451	36	≤0.001
Yes	192	81.36	24	10.17	6	2.54	14	5.93	236		
Mental and behavioral disorder											
No	31,582	91.7	1811	5.26	393	1.14	654	1.9	34,440	185.7	≤0.001
Yes	175	70.85	36	14.57	6	2.43	30	12.15	247		
Nervous system diseases											
No	31,433	91.72	1810	5.28	390	1.14	638	1.86	34,271	198.7	≤0.001
Yes	324	77.88	37	8.89	9	2.16	46	11.06	416		
Number of existing morbidities											
No co-morbidity	28,784	93.96	1269	4.14	250	0.82	330	1.08	30,633	2321.8	≤0.001
Single co-morbidity	1860	76.26	342	14.02	68	2.79	169	6.93	2439		
Multiple co-morbidity	1113	68.92	236	14.61	81	5.02	185	11.46	1615		
Total	31,757	91.55	1847	5.32	399	1.15	684	1.97	34,687		

**Table 3 ijerph-19-14381-t003:** Risk factors for COVID-19 severity using multivariable logistic regression, N = 4220.

	**Model 1**			**Model 2**			**Model 3**		
	**aOR**	**CI (95%)**	** *p* ** **-Value**	**aOR**	**CI (95%)**	** *p* ** **-Value**	**aOR**	**CI (95%)**	** *p* ** **-Value**
Chronic lower respiratory diseases									
No	1			1			1		
Yes	0.78	0.45, 1.35	0.37	0.88	0.50, 1.56	0.67	0.67	0.38, 1.19	0.17
Diabetes									
No				1			1		
Yes	2.32 *	1.87, 2.88	≤0.01	1.87 *	1.49, 2.34	≤0.01	1.3	0.95, 1.77	0.1
Thyroid gland disorders									
No				1			1		
Yes	1.61	0.99, 2.60	0.05	1.78 *	1.07, 2.97	0.03	1.35	0.80, 2.29	0.26
Hypertensive disorders									
No				1			1		
Yes	1.18	0.91, 1.52	0.22	1.04	0.80, 1.36	0.75	0.76	0.55, 1.06	0.11
Ischemic heart diseases									
No				1			1		
Yes	3.23 *	2.19, 4.76	≤0.01	2.56 *	1.72, 3.82	≤0.01	2.08 *	1.37, 3.15	≤0.01
Pulmonary and other heart diseases									
No				1			1		
Yes	2.86 *	1.88, 4.35	≤0.01	2.75 *	1.80, 4.21	≤0.01	2.13 *	1.36, 3.32	≤0.01
Cerebrovascular diseases									
No	1			1			1		
Yes	1.01	0.53, 1.91	0.98	0.9	0.48, 1.72	0.76	0.82	0.44, 1.53	0.53
Chronic kidney diseases									
No	1			1			1		
Yes	1.98 *	1.10, 3.57	0.02	2.00 *	1.09, 3.65	0.02	1.81 *	1.01, 3.25	0.05
Neoplasms									
No	1			1			1		
Yes	1.35	0.74, 2.47	0.32	1.36	0.73, 2.54	0.33	0.98	0.52, 1.87	0.96
Mental and behavioral disorder									
No	1			1			1		
Yes	1.61	0.96, 2.71	0.07	1.80 *	1.06, 3.05	0.03	1.48	0.88, 2.50	0.14
Nervous system diseases									
No	1			1			1		
Yes	1.01	0.64, 1.61	0.96	1.11	0.69, 1.79	0.66	0.82	0.50, 1.35	0.44
Number of existing morbidities									
No co-morbidity							1		
Single co-morbidity							1.52 *	1.13, 2.03	0.01
Multiple co-morbidity							2.33 *	1.37, 3.97	≤0.01
Age group									
20–29				1			1		
30–39				1.51 *	1.08, 2.11	0.02	1.51 *	1.08, 2.11	0.02
40–49				2.04 *	1.45, 2.86	≤0.01	2.01 *	1.43, 2.83	≤0.01
50–59				2.47 *	1.73, 3.52	≤0.01	2.41 *	1.69, 3.44	≤0.01
60 and above				3.28 *	2.24, 4.80	≤0.01	3.25 *	2.22, 4.76	≤0.01
Sex									
Male				1			1		
Female				0.70 *	0.55, 0.89	≤0.01	0.70 *	0.55, 0.89	≤0.01
BMI									
Normal Weight				1			1		
Underweight				2.24 *	1.18, 4.23	0.01	2.22 *	1.17, 4.21	0.01
Overweight				1.39 *	1.13, 1.72	≤0.01	1.41 *	1.14, 1.74	≤0.01
Obesity				1.58 *	1.25, 1.99	≤0.01	1.57 *	1.25, 1.98	≤0.01
Nationality									
Emirati				1			1		
Non-Emirati				1.62 *	1.16, 2.27	0.01	1.66 *	1.20, 2.31	0.01
Smoking status									
Never				1			1		
Former				1.19	0.55, 2.55	0.66	1.17	0.54, 2.52	0.69
Current				1.35	0.74, 2.46	0.33	1.34	0.73, 2.44	0.01
Pseudo R^2^	0.062			0.088			0.090		

* Indicates *p*-value < 0.05, AOR: adjusted odds ratio, Model 1—Regression with just the co-morbidities, Model 2—Regression with comorbidities and demographic characteristics, Model 3—Regression with comorbidities, demographic characteristics, and number of comorbidities.

## Data Availability

Data used in this study were retrieved from hospital inpatient records and due to data protection policies cannot be shared at individual level.

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
