# Peer review of "An Evaluation of Non-Communicable Diseases and Risk Factors Associated with COVID-19 Disease Severity in Dubai, United Arab Emirates: An Observational Retrospective Study"

_ijerph, 2022, doi:10.3390/ijerph192114381_

Round 1
Reviewer 1 Report
The paper is generally well written but the data analysis part can be substantially improved. Instead of simple analyses, more detailed or advanced analyses can be included for better conveying the information and summarizing the results.
Author Response
Reviewer #1:
Reviewer’s response to journal questions:
- English language and style are fine/minor spell check required
- Does the introduction provide sufficient background and include all relevant references? (Can be improved).
- Are all the cited references relevant to the research? (Can be improved)
- Is the research design appropriate? (Can be improved)
- Are the methods adequately described? (Can be improved)
- Are the results clearly presented? (Can be improved)
- Are the conclusions supported by the results? (Can be improved)
Author's response: The manuscript has been further revised for English language and grammatical edits by one of the authors, who is a native English speaker. We also have made revisions throughout the manuscript incorporating all reviewers’ comments which we believe have significantly improved the manuscript.
Comment#1: The paper is generally well written, but the data analysis part can be substantially improved. Instead of simple analyses, more detailed or advanced analyses can be included for better conveying the information and summarizing the results.
Author's response: We thank the reviewer for this suggestion. The aim of this paper is to determine the prevalence of non-communicable diseases and their correlation with COVID-19 disease severity among patients infected in Dubai. To achieve this objective, we have conducted both simple and advanced analyses. Our analyses included descriptive statistics, bivariate analysis (Chi-square test), and regression analysis (multiple logistic regression) using three different models. We feel that we have used the appropriate statistical tests that meet our study objectives. We also think that using other statistical tests will not add anything new to this manuscript. However, to better summarize the results, a new table was constructed to show the clinical characteristics, and Chi-square results of the 4220 patients included in the three regression models (please see Appendix 2).
Reviewer 2 Report
The authors analyzed COVID-19 hospital records from Dubai, United Arab Emirates to identify the risk factors associated with increased COVID-19 severity, using patient data between January 28th 2020 to September 30 2020. The paper is well written and has cited a very wide range of relevant research literature. Particularly interesting is that the authors have investigated underweight BMI as a risk factor for COVID-19 and discovered that it is in fact more dangerous than being overweight. Also interesting is their finding of strong association of severity of COVID-19 with number of comorbidities.
I have some suggestions for minor revisions as follows:
1. Using data from 2020 has two advantages: (1) the results are not confounded by vaccination status, since vaccines were not available during 2020; (2) because natural immunity held up throughout 2020 [1] in terms of preventing a serious second COVID reinfection, the results are also not confounded by patient status with respect to having had a previous COVID-19 infection. A disadvantage is that the virus has since mutated substantially, so there can be some quantitative variability in the reported associations for the viral strains that are now circulating during 2022. However, qualitatively speaking, the associations can be reasonably expected to persist as reported. The authors should mention the above as strengths and limitations of their data set.
2. On section 2.3, the authors should state, in terms of BMI intervals, the precise definition that they have used for: normal weight, underweight, overweight, and obese.
3. On Figure 2, the vertical axis is labeled as " length of ICU stay (days)". However, the text above the figure indicates that the numbers reported in the figure are the probability of admission to the ICU and not the length of stay in the ICU. Since the numbers are reported to 0.1 precision, I think these numbers are indeed probabilities of admission to ICU, and that the vertical axis of Figure 2 has been mislabeled. Should be easy to correct. If these are percentages, then the percent notation % should be used.
4. For Table 2, the author should explicitly state their definition for vitamin D deficiency. I am guessing that the authors define vitamin D deficiency as levels less than 20ng/ml. It may be interesting to explore, either in this paper or in future work, the sensitivity of these results to using different thresholds. For example, it is reasonable to expect some benefit by high vitamin D levels with respect to COVID-19 severity, when they exceed 50ng/ml.
5. On the first paragraph of section 4 the author's mentioned an "absence of specific antiviral treatments". It is more precise to say that COVID-19 treatment protocols have been proposed [2,3,4,5] but have not yet been widely adopted on a worldwide basis. Table 1 of Ref. [3] shows the status of worldwide adoption of treatment protocols as of December 2020, which varied between different nations. More recently, note that the antiviral medications paxlovid and molnupiravir have received an emergency use authorization (EUA) in the United States during 2022.
6. In light of the preceding comment it would be a good idea to clarify in the paper the standard of care that was given to the patients that were treated in Dubai during 2020, as context for understanding the reported outcomes, in comparison with other studies.
References
[1] Murchu, E.; Byrne, P.; Carty, P.; Gascun, C.D.; Keogan, M.; O’Neill, M.; Harrington, P.; Ryan, M. Quantifying the risk of SARS-CoV-2 reinfection over time. Rev. Med. Virol. 2022, 32, e2260
[2] Scholz, M.; Derwand, R.; Zelenko, V. COVID-19 outpatients-Early risk-stratified treatment with zinc plus low dose hydroxy- chloroquine and azithromycin: A retrospective case series study. Int. J. Antimicrob. Agents 2020, 56, 106214.
[3] McCullough, P.; Alexander, P.; Armstrong, R.; Arvinte, C.; Bain, A.; Bartlett, R.; Berkowitz, R.; Berry, A.; Borody, T.; Brewer, J.; et al. Multifaceted highly targeted sequential multidrug treatment of early ambulatory high-risk SARS-CoV-2 infection (COVID-19). Rev. Cardiovasc. Med. 2020, 21, 517–530.
[4] Marik, P.; Kory, P.; Varon, J.; Iglesias, J.; Meduri, G. MATH+ protocol for the treatment of SARS-CoV-2 infection: The scientific rationale. Expert Rev. Anti-Infect. Ther. 2021, 19, 129-135
[5] Kory, P.; Meduri, G.; Iglesias, J.; Varon, J.; Cadegiani, F.; Marik, P. MATH+ Multi-Modal Hospital Treatment Protocol for COVID-19 Infection: Clinical and Scientific Rationale. J. Clin. Med. Res. 2022, 14, 53-79.
Author Response
Reviewer #2:
Reviewer’s response to journal questions:
- English language and style are fine/minor spell check required
- Does the introduction provide sufficient background and include all relevant references? (Yes).
- Are all the cited references relevant to the research? (Yes).
- Is the research design appropriate? (Yes).
- Are the methods adequately described? (Can be improved)
- Are the results clearly presented? (Yes).
- Are the conclusions supported by the results? (Yes).
Author's response: The manuscript has been revised for English language and style by one of the authors, who is a native English speaker. We have also revised the methods section to define BMI and Vitamin D deficiency, as suggested below.
Comment#1: The authors analysed COVID-19 hospital records from Dubai, United Arab Emirates, to identify the risk factors associated with increased COVID-19 severity, using patient data between January 28th, 2020, to September 30, 2020. The paper is well written and has cited a very wide range of relevant research literature. Particularly interesting is that the authors have investigated underweight BMI as a risk factor for COVID-19 and discovered that it is in fact more dangerous than being overweight. Also interesting is their finding of strong association of severity of COVID-19 with number of comorbidities.
Author's response: We thank the reviewer for their time, constructive feedback, and support of our work.
Comment#2: Using data from 2020 has two advantages: (1) the results are not confounded by vaccination status since vaccines were not available during 2020; (2) because natural immunity held up throughout 2020 [1] in terms of preventing a serious second COVID reinfection, the results are also not confounded by patient status with respect to having had a previous COVID-19 infection. A disadvantage is that the virus has since mutated substantially, so there can be some quantitative variability in the reported associations for the viral strains that are now circulating during 2022. However, qualitatively speaking, the associations can be reasonably expected to persist as reported. The authors should mention the above as strengths and limitations of their data set.
Author's response: We thank the reviewer for the suggestion and constructive feedback. As suggested, the above-mentioned strengths and limitations were added to the revised manuscript on page 11. Please see section 4 (last two paragraphs).
Comment#3: On section 2.3, the authors should state, in terms of BMI intervals, the precise definition that they have used for: normal weight, underweight, overweight, and obese.
Author's response: We thank the reviewer for the comment. As suggested, the definitions of BMI intervals (underweight, normal, overweight, and obese) were added to the methods section 2.3.
Comment#4: On Figure 2, the vertical axis is labelled as " length of ICU stay (days)". However, the text above the figure indicates that the numbers reported in the figure are the probability of admission to the ICU and not the length of stay in the ICU. Since the numbers are reported to 0.1 precision, I think these numbers are indeed probabilities of admission to ICU, and that the vertical axis of Figure 2 has been mislabelled. Should be easy to correct. If these are percentages, then the percent notation % should be used.
Author's response: We thank the reviewer for highlighting this and have corrected the vertical axis labeled on Figure 2. Now the vertical axis label indicates percentage instead of the number of days.
Comment#5: For Table 2, the author should explicitly state their definition for vitamin D deficiency. I am guessing that the authors define vitamin D deficiency as levels less than 20ng/ml. It may be interesting to explore, either in this paper or in future work, the sensitivity of these results to using different thresholds. For example, it is reasonable to expect some benefit by high vitamin D levels with respect to COVID-19 severity, when they exceed 50ng/ml.
Author's response: The classification of Vitamin D deficiency according to the CDC and Institute of Medicine of less than 30nmol/l was used. As suggested, we have clarified this in the methods section 2.3 and Table 2. We agree with the reviewer that it would be interesting to explore the sensitivity of these results using the different thresholds however feel that it will be beyond the scope of this paper and intend to investigate this further in future work.
Comment#6: On the first paragraph of section 4 the author's mentioned an "absence of specific antiviral treatments". It is more precise to say that COVID-19 treatment protocols have been proposed [2,3,4,5] but have not yet been widely adopted on a worldwide basis. Table 1 of Ref. [3] shows the status of worldwide adoption of treatment protocols as of December 2020, which varied between different nations. More recently, note that the antiviral medications paxlovid and molnupiravir have received an emergency use authorization (EUA) in the United States during 2022.
Author's response: As suggested, the above-mentioned statement about treatment protocols have been added to the first paragraph of section 4, page 9.
Comment#7: In light of the preceding comment, it would be a good idea to clarify in the paper the standard of care that was given to the patients that were treated in Dubai during 2020, as context for understanding the reported outcomes, in comparison with other studies.
Author's response:
We thank the reviewer for this suggestion. A statement about the standard of care for patients treated in Dubai during 2020 was added to the first paragraph in section 4, page 9.
Reviewer 3 Report
This paper is well-written and the research is conducted in a good way. Despite being the theme not original, I do believe this paper is somewhat valuable, presenting the data of a large population.
Minor issue:
Can the author provide a table with the clinical characteristics of the 4220 patients whose data were used for modeling? Is the Chi-Square test for comorbidities significant also in this cohort?
Author Response
Reviewer #3:
Reviewer’s response to journal questions:
- English language and style are fine/minor spell check required
- Does the introduction provide sufficient background and include all relevant references? (Yes).
- Are all the cited references relevant to the research? (Yes).
- Is the research design appropriate? (Can be improved)
- Are the methods adequately described? (Can be improved)
- Are the results clearly presented? (Can be improved)
- Are the conclusions supported by the results? (Can be improved)
Authors’ response: We thank the reviewer for their comments and feedback. The manuscript has been further revised and edited for English by one of the authors who is a native English speaker. Further revisions to the methods, results and conclusions have also been made.
Comment#1: This paper is well-written, and the research is conducted in a good way. Despite being the theme not original, I do believe this paper is somewhat valuable, presenting the data of a large population.
Author's response: We thank the reviewer for their support of our work.
Comment#2: Minor issue: Can the author provide a table with the clinical characteristics of the 4220 patients whose data were used for modelling? Is the Chi-Square test for comorbidities significant also in this cohort?
Author's response: As advised, a new table was constructed to show the clinical characteristics and Chi-square results of the 4220 patients. Please see Appendix 2. As you will find, the number of existing co-morbidities was significantly associated with COVID-19 severity.